# Pre-Harvest Foliar Application of Mineral Nutrients to Retard Chlorophyll Degradation and Preserve Bio-Active Compounds in Broccoli

**Mohamed M. El-Mogy [1],\*** , **Abdel Wahab M. Mahmoud [2]** , **Mohamed B. I. El-Sawy [3]** **and Aditya Parmar [4]**

[1] Vegetable Crops Department, Faculty of Agriculture, Cairo University, 12613 Giza, Egypt
[2] Plant Physiology Department, Faculty of Agriculture, Cairo University, 12613 Giza, Egypt; mohamed.mahmoud@agr.cu.edu.eg
[3] Horticulture Department, Faculty of Agriculture, Kafrelsheikh University, 33511 Kafrelsheikh, Egypt; drmohamedelsawy@yahoo.com
[4] Natural Resources Institute, University of Greenwich, Chatham ME44TB, UK; A.Parmar@greenwich.ac.uk
\* Correspondence: elmogy@agr.cu.edu.eg; Tel.: +002-010-6802-7607; Fax: +02-3571-7355

**Abstract:** Foliar application of micronutrients has become a common farm management practice to increase the overall yield of various crops. However, the effects of foliar fertilization on shelf life and postharvest quality of the crops are rather under-researched. The aim of this field experiment was to evaluate the effect of foliar application of individual mineral nutrients (calcium (Ca), zinc (Zn), manganese (Mn), and iron (Fe) on pre and postharvest quality of broccoli. The broccoli plants were subjected to single foliar sprays of either Ca, Fe, Zn, or Mn, which was repeated four times during plant growth at a 1 g/kg concentration. Once harvested, the broccoli heads were refrigerated at 4 °C for 28 days. Our results indicated that foliar application of Ca, Zn, Mn, and Fe did not have a significant effect on plant growth parameters, apart from enhancing Soil Plant Analysis Development (SPAD) chlorophyll meter values. However, during postharvest, foliar application treatment showed a positive response on weight loss during storage and reduction in yellowing of the broccoli heads. Foliar treatments increased the concentration of Nitrogen (N), Phosphorus (P), Ca, Zn, Mn and Fe significantly in the broccoli head tissue. Total chlorophyll content, total phenolic compound, ascorbic acid, peroxidase activity, glucoraphanin and glucobrassicin and flavonoids were significantly increased by all foliar treatments. Crude protein content and sulforaphane were enhanced by Ca and Mn treatments. Overall, foliar application of the investigated mineral nutrients may prove beneficial in improving the shelf-life and nutrient content of broccoli during postharvest handling and storage.

**Keywords:** foliar application; mineral nutrients; Ca; plant growth; postharvest quality; chlorophyll

## 1. Introduction

Broccoli (*Brassica oleracea* var. *Italica*) is a dicotyledonous biennial herbaceous member of cabbage family Brassicaceae, genus Brassica, belonging to the larger group of Cruciferae [1]. Other known vegetables that belong to this family are brussels sprouts, cabbage, cauliflower, Chinese cabbage, kale, and kohlrabi. The green colour of broccoli comes from the chlorophyll present in sepals of its floral buds [2]. Apart from being a high commercial value crop, broccoli is highly nutritious and health-promoting food, due to its high vitamin (A, C, E, B2 and K1), minerals (Ca, Na, Mg, K, Zn, Fe, etc.) and phytochemical content (primarily glucosinolates and phenolic antioxidants).

One of the main postharvest problems of this crop is chlorophyll degradation and rapid senescence of broccoli heads [3]. Various post-harvest treatments such as edible coatings and γ-irradiation [4],

modified atmosphere packaging (MAP) [5], exogenous sodium nitroprusside treatment [3], and hot water and sucrose treatment [6,7] has been tested fairly recently. However, research on pre-harvest treatments to improve postharvest quality of broccoli is rather limited. Foliar application has been termed as the most efficient way of fertilization due to its rapid and efficient response to the plant, requirement of less product and independence of soil conditions [8]. Majority of the past studies looked yield potential increase through foliar application of micronutrient, hence the studies focusing on postharvest shelf life increase are rather limited.

This study focuses on the foliar application of four important mineral nutrients, namely calcium (Ca), zinc (Zn), manganese (Mn), and iron (Fe). Ca is one the most important mineral, essential for maintaining the quality shelf-life of horticultural produce, primarily due to its role in preserving the cell membrane integrity [9]. Zn is one of the most critical micronutrients for growth and development in plants through the formation of indole-tic acid [10]. Zn deficiency in alkaline and sandy soils is a severe problem, situation is exacerbated further by excessive use of $P_2O_5$ fertilizer [11]. Mn is important micronutrient in plants responsible for chlorophyll formation, enzymatic respiration, germination, maturity, and disease resistance [12–14]. Fe deficiency is a common problem in horticultural production particularly in alkaline soils, due to poor Fe mineral absorption rates [15,16]. Foliar application of Fe has shown positive effects on yield of tomatoes and pepper [13,17,18]. Fe is of interest regarding broccoli heads, as there is limited knowledge on Fe foliar application and broccoli inflorescence.

To our knowledge, there is a significant knowledge gap on the influence of foliar fertilization of broccoli heads by Ca, Zn, Mn, and Fe on growth, bioactive compounds, quality, and chlorophyll degradation of broccoli heads. We hypothesize that under Mediterranean semi-arid regions (alkaline soils) the foliar application of Ca, Zn, Mn, and Fe will increase broccoli heads quality and retard senescence and yellowing during refrigerated storage.

## 2. Materials and Methods

### 2.1. Plant Material and Treatments

The field experiment was conducted during a period of two years (2016–2017 and 2017–2018) in winter seasons (October 2016–March 2017 and October 2017 –March 2018). Broccoli cultivar 'Imperial (F1)' was used for experiment. The experimental design was randomized complete block with four replicates. Each experimental plot was 11.2 m$^2$ with four rows (4 m length and 0.7 m width), located between two guard rows. The soil texture was clay-loamy with 30, 35 and 35% sand, silt, and clay content respectively. The chemical properties of experimental soil were pH: 7.82, ECe: (dS m$^{-1}$): 2.48, N (mg kg$^{-1}$): 40.34, P (mg kg$^{-1}$): 57.13, K (mg kg$^{-1}$): 489, Fe (mg kg$^{-1}$): 4.73, Mn (mg kg$^{-1}$): 17.34. The maximum ambient temperature during growing season from October to March was ranging from 20 to 29 °C in 2016–2017 and from 18 to 27 °C in 2017–2018, while the minimum temperature ranged from 5 to 16 °C and from 7 to 19 °C in the first and second growing seasons, respectively. Also, the total rainfall in both years was < 22 mm. Seedling of broccoli was transplanted at 0.5 m from each other in rows. Weeds were removed manually after emergence. All treatments received 92 kg nitrogen (N, Urea), 96 kg $P_2O_5$ (super phosphate), and 90 kg $K_2O$ ha$^{-1}$ (potassium sulphate). Foliar application of Ca (Calcium 10%, Union calcium, AGAS, Cairo, Egypt), Zn (EDTA 13%, Techno Green, Cairo, Egypt), Mn (EDTA 13%, Techno Green, Cairo, Egypt), and Fe (Fe-EDTA 13%, AGAS, Cairo, Egypt) was carried out at a rate of 1 g·L$^{-1}$, by a manual knapsack sprayer. The application concentration was selected based on preliminary test conducted at the university experimental farm. The application was begun after 30 days from transplanting and repeated four times at 10-day intervals.

### 2.2. Determination of Plant Growth

Ten broccoli plants from every experimental plot were randomly harvested at commercial maturity (head reached to its maximum size and compacted). Total plant fresh weight, broccoli head weight,

head diameter, number of leaves, root weight and leaves the dry matter, and Soil Plant Analyses Development (SPAD) by SPAD meter (SPAD-502 Minolta Co, Osaka, Japan) values were recorded.

## 2.3. Determination of Postharvest Quality

Healthy broccoli heads (free from damage and defect) were separated from the plant by a sharp knife and transported (within two hours) to the postharvest laboratory at Faculty of Agriculture, Cairo University. Each head was wrapped in shrink warp, and packed in carboard boxed before refrigerated storage at 4 °C and 90% RH. The weight loss (expressed in % weight loss) and head surface colour (Chroma Meter (CR-400, Konica Minolta, Tokyo, Japan) were measured at an interval of 7 days for a period of 28 days. Head surface colour was measured for 10 heads per replicate to determine the changes in green colour of broccoli heads. L* (lightness), chroma, and hue angle were determined. Each measurement was taken at three locations for each individual head. After 28 of storage, the colour of stem surface was taken to identify the changes of white colour during storage.

## 2.4. Chemicals and Bioactive Compounds

### 2.4.1. Nitrogen, Crude Protein, Phosphorus, Potassium, and Vitamin C

Modified- micro-Kjeldahl method was used for Nitrogen (N) and crude protein (N × 6.25) determination according to Helrich [19]. Phosphorus (P) content was determined according to Jackson [20]. Potassium content (K) was determined by using the flame photometer apparatus (CORNING M 410, Essex, UK). Vitamin C (ascorbic acid) content was determined as per the method presented by Helrich [19].

### 2.4.2. Calcium, Iron, Manganese, and Zinc Content

Ca, Fe, Mn, and Zn content were determined using Atomic Absorption Spectrophotometer (Pye Unicam, model SP-1900, Cambridge, UK) with air-acetylene fuel. In briefly, the wet digestion of 0.2 g plant material (broccoli head) with sulphuric and perchloric acids was carried out by adding concentrated sulphuric acid (5 mL) to the samples and the mixture was heated for 10 min. Then 0.5 mL perchloric acid was added and heated continuously till a clear solution was obtained. The digested solution was quantitatively transferred to a 100 mL volumetric flask using deionized water according to Helrich [19] for analysis

### 2.4.3. Total Chlorophylls, Phenolic Contents, Total Flavonoids

Total chlorophyll content of fresh broccoli heads was measured according to Moran [21]. Total phenolic contents (TPC) were determined by the method presented by Singleton and Rossi [22] using Folin–Ciocalteu colourimetric method. Total flavonoids were determined by the method described by Meda et al [23].

### 2.4.4. Sulforaphane Extraction

Sulforaphane extraction was conducted as per the method described by Gu et al. [24] and Han and Row [25]. In this method 0.2 g of freeze-dried samples subjected to serial dilution (1:20, 1:30, 1:40, 1:50 and 1:60) with acidic water (pH 3.0). The samples were then incubated at 50, 55, 60, 65 and 70 °C for 1, 2, 3, 4 and 5 h. Once incubated, 40 mL of dichloromethane was added, and the mixture was well vortexed and filtered through 0.45 mL membrane. HPLC (High-Performance Liquid Chromatography) analyses were carried out by injecting 20 mL aliquot on a Waters E2695 Liquid Chromatograph (Waters Crop., Milford, MA, USA) connected to a model 2998 (PAD) photodiode array detector.

### 2.4.5. Extraction of Glucosinolates

Glucosinolates were extracted according to Bjerg et al. [26] and Bjerg and Sorensen [27] from freeze-dried broccoli powder. Glucobarbarin as internal standard was used for calculating the

glucosinolate concentration. In this method, 0.2 g of sample was spiked with a 100 μl standard solution containing 5.0 μmol·mL$^{-1}$ of sinigrin and glucobarbarin. The resulting mixed was then homogenized three times with 5 mL of boiling methanol (70%) for two min using an Ultra-Turrax Homogenizer (Ika-Labortechnik, Staufen, Germany). Once homogenized, the samples were centrifuged and concentrated to dryness in vacuo. The residue was dissolved in 2 mL deionized water for HPLC analysis of Glucosinolates

### 2.4.6. Peroxidase Activity

Fresh broccoli samples (0.5 g) were frozen in liquid nitrogen to extract the peroxidase enzyme. The samples were grinded with 10 mL extraction buffer (50 mM phosphate buffer, pH 7, containing 0.5 mM EDTA and 2% PVPP (w/v)) and centrifuged at 21925 rpm for 20 min. The resultant supernatant was used to determine peroxidase activity as mentioned by In et al. [28] which prescribe a spectrophotometric method by the formation of guaiacol in l mL reaction mixture (450 μl 25 mM guaiacol, 450 μl 225 mM H$_2$O$_2$) and 100 μl crude enzymes. The increase in absorbance was recorded by the addition of H$_2$O$_2$ at 470 nm for 2 min (e, 26.6 mM$^{-1}$ cm$^{-1}$).

### 2.5. Statistical Analysis

The field experiment was arranged in Randomized Complete Block Design. The experimental data were expressed as the mean and standard error (SE) with four replicates ($n = 4$). The SPSS 20.0 (SPSS Inc., Chicago, IL, USA) software was used for testing significant differences using Tukey's multiple range tests at $p < 0.05$.

## 3. Results and Discussion

### 3.1. Plant Growth and SPAD

Plant growth parameters (plant fresh weight, number of leaves, head weight, head diameter, root weight, and dry weight of leaves) were not influenced by foliar application in both seasons. SPAD readings were increased significantly ($p < 0.05$) by all treatments in 2017 (Figure 1A,B). In 2018, although SPAD values were slightly higher for all treatments, however, only Zn application showed significantly elevated value in comparison to control. In general, SPAD meter values are proportional to the amount of chlorophyll present in plant tissue and serves as a good nondestructive indicator of chlorophyll. This can be observed by relating the total chlorophyll content results at harvest (0 d) in Table 3 to SPAD values. In 2017 and 2018, all the treatment showed a higher chlorophyll content at harvest, like the SPAD reading.

Increased SPAD values by Zn treatment might be due to its function in enhancing the enzymes (aminolevulinic acid dehydratase (ALA-dehydratase)) responsible for chlorophyll biosynthesis [29,30]. Similarly, the effect of Fe and Mn in increasing SPAD reading values could be due to their role in chlorophyll synthesis and photosynthesis process [31]. The increase in SPAD due to Ca applicable has been previously attributed to its role in controlling the osmotic strength of the cytoplasm which is critical in preventing cell dehydration [32].

### 3.2. Weight Loss

Weight loss in the form of loss of moisture due to dehydration and respiration activities in one of the most important postharvest issue which has a direct impact on marketability and quality of the produce [33]. In this study, we observed that weight loss in broccoli heads gradually increased during storage periods for both years. All the treatments resulted in lower weight loss in comparison to control (5.6%) as shown in Figure 1C,D. At the end of storage period, the minimum weight loss (2.9%) was recorded for Ca application, followed by Mn (3.1 %), Fe (3.3 %), and Zn (4.8 %). These trends were observed for both the year 2017 and 2018.

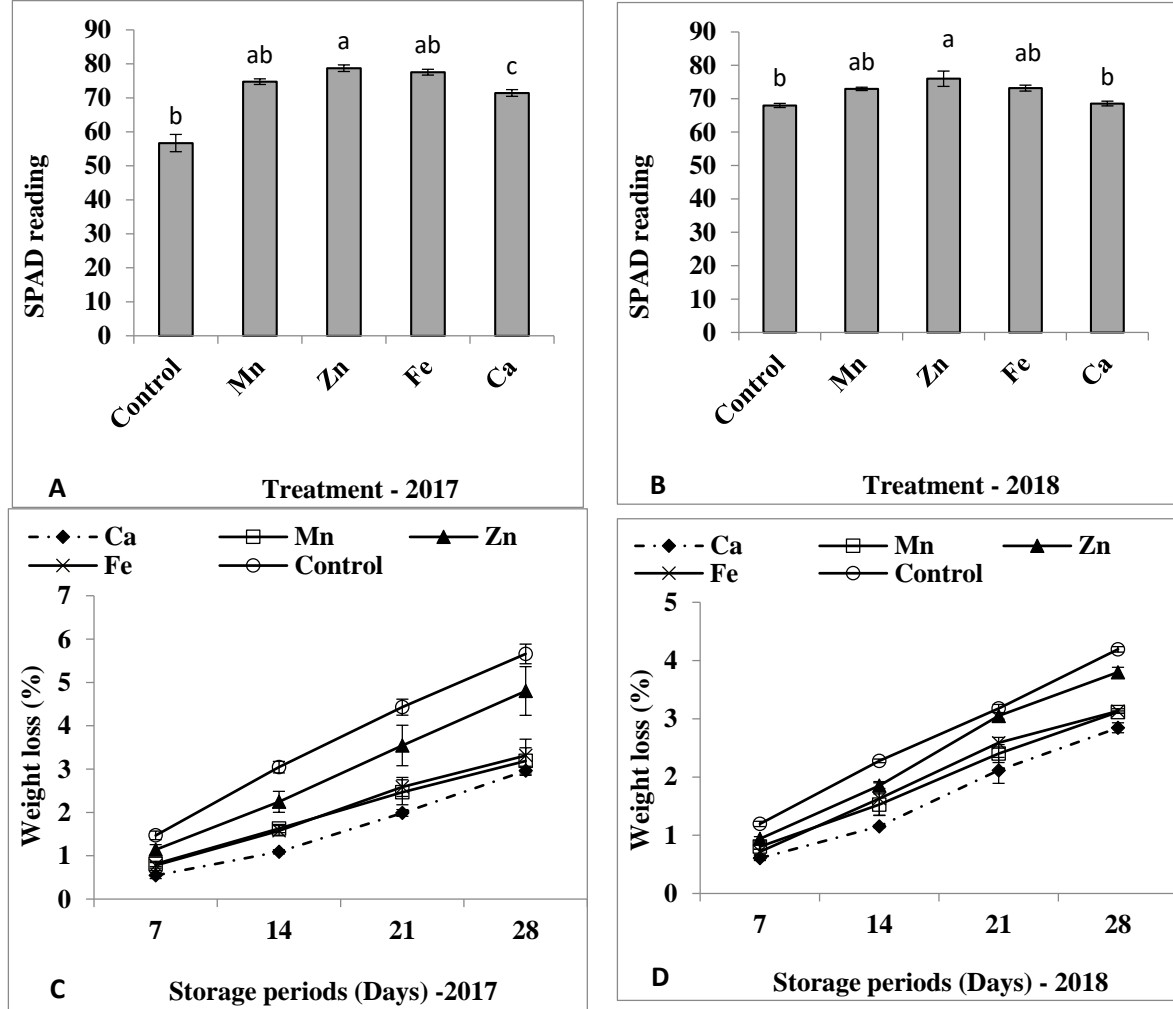

**Figure 1.** Effect of foliar application with Ca, Mn, Zn, and Fe elements on (**A**) Soil Plant Analysis Development (SPAD) reading of broccoli leaves at harvest stage in 2017, (**B**) SPAD reading in 2018, (**C**) weight loss of broccoli heads stored for 28 days at 4 °C in 2017, and (**D**) weight loss in 2018. Different letters indicate significant differences between treatments (Tukey test at $p < 0.05$).

The positive effects of Ca in retarding weight loss of broccoli is presumably due to its importance in maintaining cell wall structure and firmness [34]. Duffy [35] reported that Zn application improves water use efficiency and turgidity in plant tissue, which could be responsible for maintaining higher moisture levels inside broccoli heads tissues. Moreover, it is known that enhanced Mn concentration in plant tissue increases the synthesis of secondary metabolites, which produce lignin [36]. Hence, an increase in lignin compound due to higher Mn levels in broccoli tissue could be responsible for minimize water loss in our experiment. Fe as a micronutrient has an important role to play in photosynthesis process [37] and has properties such as being an osmoprotectant, nitrogen assimilator, and defence against pathogens [38].

### 3.3. Surface Colour

The green colour of broccoli heads is the most important quality trait which determines its marketability and economic value. In this study, we found that changes in hue angle, $L^*$, and chroma value were significantly influenced by the Mn and Zn foliar application treatments (Figure 2). The data in Figure 2A,B indicates that higher hue angle values which mean less yellow and greener colour was recorded in Mn treated broccoli heads, followed by Zn. Whereas, no significant differences

were found between Fe and Ca treatments. The control plants showed the lowest hue angle values. Also, the lowest L* values which mean less lightness and more darkness was recorded in Mn and Zn treatments followed by Fe and Ca treatments (Figure 2C,D). Similar trend was observed also in chroma values (Figure 2E,F). Mn plays an important role in photosynthesis process [31], which results in an increase in colour density. The positive effects of Zn increasing green colour of broccoli heads are probably due to its role in increasing chlorophyll content and nett photosynthetic rate [39]. Our results are in agreement with Mohammadi et al. [40] who stated that Fe application enhanced photosynthetic pigments of peppermint. According to Gu et al. [41], chlorophyll content and leaf photosynthetic rate of peanut plants were increased by Ca Fertilizer.

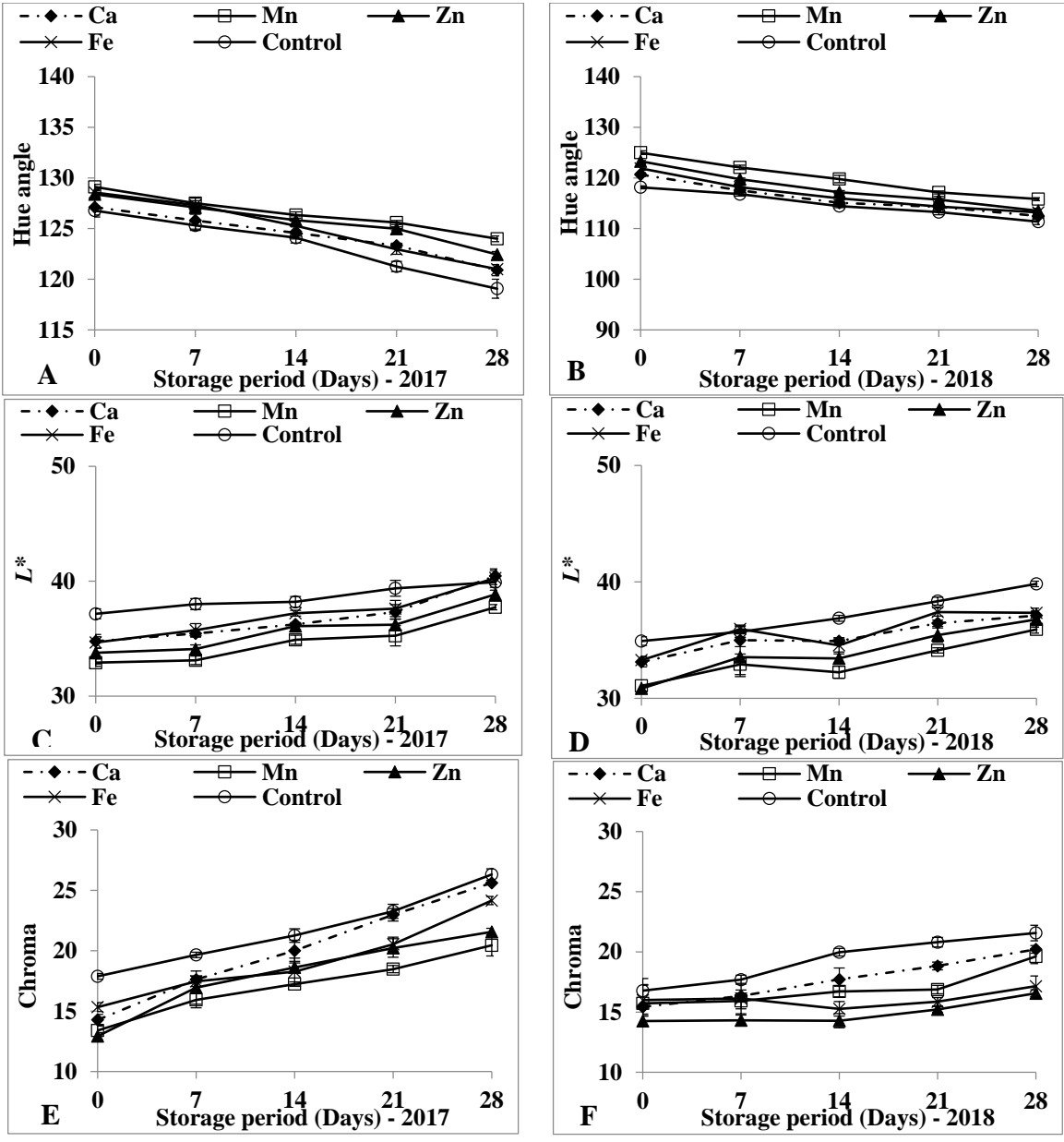

**Figure 2.** Effect of foliar application with Ca, Mn, Zn, and Fe elements on (**A**) Hue angle in 2017, (**B**) Hue angle in 2018, (**C**) L* values in 2017, (**D**) L* values in 2018, (**E**) Chroma in 2017, and (**F**) Chroma in 2018 of broccoli florets stored for 28 days at 4 °C. Different letters indicate significant differences between treatments (Tukey test at $p < 0.05$).

### 3.4. Mineral Elements

N and P contents of broccoli heads were significantly increased with Ca, Mn, Zn, and Fe foliar application in comparison to control samples in both seasons (Table 1). However, K content was not affected, which ranged from 3.53 to 3.60 % (data not shown). An increase of N content by Ca and Zn application could be attributed to their role in the metabolism process of N in plants [37]. Moreover, Mn has an important role in activating enzymes which are responsible for the biosynthesis of N and P [42].

**Table 1.** Effect of foliar application of Ca, Mn, Zn, Fe elements plus water as a control on nitrogen, phosphor, potassium, iron, manganese, zinc, and calcium content of broccoli at harvest time.

| 2017 | Cont. | Mn | Zn | Fe | Ca |
|---|---|---|---|---|---|
| N (%) | 3.37 ± 0.03 [z] c [y] | 3.79 ± 0.02 ab | 3.76 ± 0.05 ab | 3.71 ± 0.03 b | 3.94 ± 0.03 a |
| P (%) | 0.45 ± 0.02 b | 0.66 ± 0.01 a | 0.63 ± 0.02 a | 0.61 ± 0.02 a | 0.61 ± 0.01 a |
| Fe (mg/kg) | 152.65 ± 0.90 b | 152.77 ± 0.66 b | 145.48 ± 0.30 c | 165.03 ± 0.24 a | 151.25 ± 0.40 b |
| Mn (mg/kg) | 60.36 ± 0.47 b | 63.69 ± 0.27 a | 60.06 ± 0.14 bc | 59.01 ± 0.58 bc | 58.22 ± 0.59 c |
| Zn (mg/kg) | 132.67 ± 1.48 b | 131.86 ± 1.12 b | 136.29 ± 0.40 a | 132.86 ± 1.16 b | 129.26 ± 0.90 b |
| Ca (%) | 1.86 ± 0.02 b | 1.88 ± 0.03 b | 1.86 ± 0.05 b | 1.85 ± 0.04 b | 2.11 ± 0.06 a |
| **2018** | Cont. | Mn | Zn | Fe | Ca |
| N (%) | 4.20 ± 0.04 c | 4.59 ± 0.02 b | 4.58 ± 0.02 b | 4.63 ± 0.05 b | 4.84 ± 0.06 a |
| P (%) | 0.52 ± 0.07 b | 0.75 ± 0.01 a | 0.71 ± 0.01 a | 0.69 ± 0.05 ab | 0.73 ± 0.02 a |
| Fe (mg/kg) | 162.65 ± 0.91 b | 161.11 ± 1.38 b | 163.15 ± 2.35 b | 171.37 ± 0.88 a | 159.25 ± 0.80 b |
| Mn (mg/kg) | 50.36 ± 0.47 b | 54.69 ± 0.95 a | 50.73 ± 0.52 b | 51.34 ± 0.62 ab | 48.88 ± 1.20 b |
| Zn (mg/kg) | 116.34 ± 0.55 b | 117.20 ± 0.60 b | 121.39 ± 0.77 a | 114.33 ± 0.62 b | 116.59 ± 0.50 b |
| Ca (%) | 2.76 ± 0.06 b | 2.74 ± 0.06 b | 2.76 ± 0.05 b | 2.72 ± 0.06 b | 3.21 ± 0.11 a |

[z] The results are expressed as mean ± SE of four replicates followed by a letter. [y] Different letters indicate significant statistical differences among fertilizer treatments in the same row ($p < 0.05$).

As far as the individual elemental concentration of Ca, Mn, Zn and Fe is a concern, it was found that highest concentration of individual element was present in the samples which were treated with that particular element (refer to Table 1) for both the years. These results are in good agreement with Carrasco-Gil et al. [13], who reported that foliar treatment of individual elements would increase the concentration of elements such as Fe and Zn in plants. Moreover, similar results have been reported for pre-harvest foliar application of Zn and Ca elements [43,44].

### 3.5. Bioactive Compounds

Total phenolic compounds (TPC) in broccoli heads increased with increasing storage time for both seasons as shown in Table 2. A gradual increase in TPC during storage has been observed previously by Xu et al. [7], who attributed it to the induction of, phenylalanine ammonia-lyase (PAL) activity. By the end of the storage period, all the samples with foliar application of Ca, Mn, Zn and Fe had higher levels of TPC in comparison to control samples. The most prominent effect was recorded for Ca, Mn and Zn treatments as highlighted in Table 2. TPC is a highly effective antioxidant which prevents the spread of free radicals, hence, preserving the quality of fresh produce [1]. Various previous studies have shown similar trends where higher levels of TPC were recorded with Zn and Mn foliar application [45,46]. Table 2 also presents the concentration of AsA in the treatment and control samples, it is clear from these results that foliar application did not have much effect on initial concentration of the AsA in the broccoli tissue, and neither it played any significant role in retarding its degradation. Only in first season (2017), Mn application treatment seems to have retained significant higher levels of AsA, however, this was not replicated in the next season. Moreover, significant natural degradation of AsA was observed for all the samples. This natural degradation of AsA in fresh fruits and vegetables during postharvest storage is common phenomenon [47]. Hence, postharvest practitioners look for ways to retard the natural degradation of Asa to improve the final quality of the produce. Previously,

a higher level of AsA retention with pre-harvest Ca and Zn application has been reported, respectively, however in our study the level of retention was not at significant level [1,48].

**Table 2.** Effect of foliar application of Ca, Mn, Zn, Fe elements plus water as the control on the total phenolic compound (TPC) and ascorbic acid (AsA) of broccoli stored for 28 days at 4 °C. FW = fresh weight, GAE = gallic acid equivalent

| TPC Content (mg/GAE/g FW) | | | | | |
|---|---|---|---|---|---|
| **2017** | **Cont.** | **Mn** | **Zn** | **Fe** | **Ca** |
| 0 d | 90.16 ± 0.45 [z] B [y] e | 91.73 ± 0.30 AB d | 92.72 ± 0.40 A e | 91.16 ± 0.32 AB e | 90.49 ± 0.58 B e |
| 7 d | 97.923 ± 0.13 B d | 101.41 ± 0.67 A c | 96.99 ± 0.26 B d | 101.33 ± 0.72 A d | 99.00 ± 0.57 AB d |
| 14 d | 121.42 ± 0.46 A c | 120.18 ± 0.24 A b | 120.87 ± 0.34 A c | 119.42 ± 0.57 A c | 108.41 ± 1.17 B c |
| 21 d | 139.37 ± 0.74 A a | 137.22 ± 0.61 AB a | 132.30 ± 0.37 B a | 135.18 ± 0.30 AB b | 117.42 ± 3.13 C b |
| 28 d | 135.44 ± 0.63 A b | 135.61 ± 0.56 A a | 128.28 ± 0.46 B b | 131.70 ± 0.91 AB a | 122.98 ± 1.57 C a |
| **2018** | **Cont.** | **Mn** | **Zn** | **Fe** | **Ca** |
| 0 d | 95.18 ± 0.83 AB a | 97.40 ± 0.59 A e | 97.39 ± 0.69 A e | 96.83 ± 0.91 A e | 92.49 ± 0.60 B e |
| 7 d | 103.43 ± 1.18 AB b | 106.74 ± 0.41 A d | 104.05 ± 0.91 AB d | 102.71 ± 0.50 B d | 103.08 ± 0.63 B d |
| 14 d | 112.79 ± 0.93 BC c | 116.10 ± 0.69 A c | 116.54 ± 0.54 A c | 115.09 ± 0.41 AB c | 111.41 ± 0.69 C c |
| 21 d | 128.98 ± 0.35 A d | 128.21 ± 0.95 A b | 129.03 ± 0.28 A b | 125.85 ± 0.85 AB b | 124.09 ± 1.17 B b |
| 28 d | 145.44 ± 0.63 AB e | 144.43 ± 0.34 AB a | 147.81 ± 0.58 A a | 142.70 ± 0.62 B a | 143.16 ± 1.40 B a |
| AsA Content (mg/100g FW) | | | | | |
| **2017** | **Cont.** | **Mn** | **Zn** | **Fe** | **Ca** |
| 0 d | 127.27 ± 0.41 B a | 130.52 ± 0.56 A a | 132.62 ± 0.68 A a | 130.82 ± 0.18 A a | 125.65 ± 1.04 B a |
| 7 d | 123.33 ± 0.88 A b | 124.33 ± 0.33 A b | 123.54 ± 0.64 A b | 124.56 ± 0.62 A b | 121.78 ± 1.02 A b |
| 14 d | 114.59 ± 0.45 AB c | 118.79 ± 0.80 A c | 117.14 ± 0.89 A c | 117.20 ± 1.17 A c | 111.59 ± 1.82 B c |
| 21 d | 91.94 ± 1.04 BC d | 95.51 ± 0.63 AB d | 99.25 ± 0.43 A d | 97.34 ± 1.55 A d | 90.00 ± 0.92 C d |
| 28 d | 79.40 ± 0.59 C e | 95.30 ± 0.51 A d | 80.53 ± 0.48 BC e | 82.38 ± 0.62 B e | 69.92 ± 0.71 D e |
| **2018** | **Cont.** | **Mn** | **Zn** | **Fe** | **Ca** |
| 0 d | 147.96 ± 0.81 A a | 143.83 ± 1.26 AB a | 145.63 ± 1.25 AB a | 141.51 ± 0.60 B a | 143.83 ± 1.90 AB a |
| 7 d | 140.53 ± 0.54 A b | 137.03 ± 1.01 ABC b | 138.55 ± 0.63 AB b | 136.56 ± 0.65 BC a | 134.94 ± 0.84 C b |
| 14 d | 131.62 ± 0.45 A c | 128.13 ± 0.49 BC c | 131.48 ± 0.59 AB c | 126.35 ± 0.64 C b | 127.14 ± 1.27 C c |
| 21 d | 113.08 ± 1.23 A d | 100.37 ± 0.55 B d | 112.84 ± 0.55 A d | 96.81 ± 3.74 B c | 98.33 ± 0.58 B d |
| 28 d | 91.76 ± 0.87 A e | 89.36 ± 0.56 ABC e | 91.53 ± 0.71 AB e | 88.38 ± 0.56 C d | 88.59 ± 0.61 BC e |

[z] The results are expressed as mean ± SE of four replicates followed by a letter. [y] Different letters indicate significant statistical differences among fertilizer treatments (upper case) in the same row and days of storage (lower case) in the same column ($p < 0.05$).

Table 3 presents the results for crude protein (which is dependent on the N level) and total chlorophyll content. Crude protein content significantly decreased during the storage period for both the seasons. Again, like AsA, no stark difference was observed in the control and treatment samples for crude protein. There was slightly higher crude protein retention in treatment with Mn and Ca application for year 2017 and 2018 respectively, however this trend was not replicated for both years, which undermines the conclusiveness of these results.

All foliar application treatments significantly increased the chlorophyll content of broccoli heads in comparison to control at harvest (0 days) and after 21 and 28 days of storage in the first season. However, the total general trend shows degradation of chlorophyll during storage, which is in accordance with previous study from Serrano et al [47]. In 2018, total chlorophyll content was significantly higher for all the treatments at harvest time, 7- and 14-days of storage. Our results in Figure 2A,B support our hypothesis that a higher hue angle is an indicator of greenness of broccoli, which means higher chlorophyll content. Total chlorophyll content measurement is a standard method to determine the senescence process which is related to yellowing of the broccoli heads. The results for total chlorophyll content support our hypothesis that pre-harvest foliar application of Ca and Mn could help retain the chlorophyll in broccoli for a longer duration during postharvest cold storage. Guo et al. [49] also reported similar results, where application of $CaCl_2$ reduced the decline of total chlorophyll content

during storage. Mn, Zn and Fe are known to play an important role in the formation of chlorophyll in higher plants [37,39,42,50] which was indicative from our results.

**Table 3.** Effect of foliar application of Ca, Mn, Zn, Fe elements plus water as the control on crude protein and total chlorophyll of broccoli florets stored for 28 days at 4 °C.

| | Cont. | Mn | Zn | Fe | Ca |
|---|---|---|---|---|---|
| **Crude Protein Content (%)** | | | | | |
| **2017** | Cont. | Mn | Zn | Fe | Ca |
| 0 d | 20.79 ± 0.43 $^z$ C $^y$ a | 24.55 ± 0.49 AB a | 22.67 ± 0.15 BC a | 22.93 ± 0.46 B a | 25.49 ± 0.40 A a |
| 7 d | 20.54 ± 0.66 A a | 21.71 ± 0.43 A b | 23.35 ± 1.21 A a | 22.50 ± 0.69 A a | 21.86 ± 0.34 A b |
| 14 d | 12.36 ± 0.32 A b | 13.84 ± 0.49 A c | 13.33 ± 0.24 A b | 13.91 ± 0.33 A b | 13.93 ± 0.43 A c |
| 21 d | 11.68 ± 0.28 C bc | 13.02 ± 0.12 AB c | 13.16 ± 1.16 AB bc | 13.62 ± 0.23 A b | 12.33 ± 0.33 BC c |
| 28 d | 9.79 ± 0.36 A c | 10.50 ± 0.58 A d | 10.35 ± 0.53 A c | 10.20 ± 0.42 A c | 9.98 ± 0.25 A d |
| **2018** | Cont. | Mn | Zn | Fe | Ca |
| 0 d | 30.45 ± 0.58 C a | 35.89 ± 0.36 AB a | 32.67 ± 0.16 C a | 32.93 ± 0.46 BC a | 36.43 ± 1.27 A a |
| 7 d | 25.89 ± 0.76 B b | 30.04 ± 0.60 A b | 26.02 ± 0.47 B b | 26.67 ± 0.57 AB b | 27.93 ± 1.38 AB a |
| 14 d | 20.91 ± 0.80 B c | 25.51 ± 0.52 A c | 21.16 ± 1.05 B c | 22.24 ± 1.22 AB c | 20.60 ± 0.92 B b |
| 21 d | 19.45 ± 0.61 AB c | 21.55 ± 0.61 A d | 19.11 ± 0.94 AB c | 19.52 ± 0.62 AB c | 17.20 ± 0.92 B bc |
| 28 d | 9.79 ± 1.00 A d | 10.55 ± 0.65 A e | 10.98 ± 0.32 A d | 9.73 ± 0.38 A d | 19.36 ± 0.57 A c |
| **Total chlorophyll content (mg/g FW)** | | | | | |
| **2017** | Cont. | Mn | Zn | Fe | Ca |
| 0 d | 1.67 ± 0.07 B a | 1.79 ± 0.05 A a | 1.91 ± 0.03 A a | 1.91 ± 0.04 A a | 1.77 ± 0.02 A a |
| 7 d | 1.57 ± 0.05 A ab | 1.65 ± 0.10 A ab | 1.73 ± 0.08 A a | 1.70 ± 0.10 A bc | 1.65 ± 0.07 A ab |
| 14 d | 1.32 ± 0.05 A b | 1.32 ± 0.04 A c | 1.38 ± 0.02 A b | 1.46 ± 0.04 A cd | 1.29 ± 0.00 A c |
| 21 d | 1.00 ± 0.06 B c | 1.37 ± 0.09 A bc | 1.34 ± 0.07 AB b | 1.34 ± 0.05 A d | 1.34 ± 0.07 A bc |
| 28 d | 0.81 ± 0.04 B c | 0.99 ± 0.02 A d | 1.00 ± 0.03 A c | 1.00 ± 0.02 A e | 0.98 ± 0.05 A d |
| **2018** | Cont. | Mn | Zn | Fe | Ca |
| 0 d | 2.16 ± 0.09 C a | 2.74 ± 0.04 A a | 2.44 ± 0.05 AB a | 2.47 ± 0.04 AB a | 2.27 ± 0.07 BC a |
| 7 d | 1.96 ± 0.04 C b | 2.45 ± 0.06 A b | 2.20 ± 0.03 B b | 2.21 ± 0.02 B ab | 2.15 ± 0.03 B a |
| 14 d | 1.62 ± 0.05 C b | 2.13 ± 0.06 A c | 1.91 ± 0.03 B c | 1.89 ± 0.02 B bc | 1.85 ± 0.02 B b |
| 21 d | 1.51 ± 0.01 C bc | 1.90 ± 0.03 A d | 1.64 ± 0.05 BC d | 1.71 ± 0.04 B c | 1.60 ± 0.03 BC c |
| 28 d | 1.27 ± 0.06 A c | 1.49 ± 0.04 A e | 1.27 ± 0.04 A e | 1.44 ± 0.06 A d | 1.36 ± 0.05 A d |

$^z$ The results are expressed as mean ± SE of four replicates followed by a letter. $^y$ Different letters indicate significant statistical differences among fertilizer treatments (upper case) in the same row and days of storage (lower case) in the same column ($p < 0.05$).

Glucoraphanin and glucobrassicin contents of broccoli heads were linearly significantly decreased with increasing storage durations in both seasons (Table 4). Foliar application treatments with Ca, Fe, Mn, and Zn significantly increased glucoraphanin content in comparison to control treatment during whole storage periods. In addition, in both seasons, the higher values of glucobrassicin content were obtained from Mn, Zn, and Fe treatments during storage. Glucosinolates are considered as one of the most bioactive compounds endure in the plants which play an important role in defence against plant pathogens [1]. Glucoraphanin is the precursor of sulforaphane which is converted by myrosinase endogenous enzyme [51]. The previous studies have reported a decrease in glucoraphanin content of broccoli heads after refrigerated storage at 4 °C [52]. Our result showed that pre-harvest Ca treatment enhanced glucoraphanin content and prevented the reduction during storage. Similarly, Sun et al. [53] reported an increase of glucoraphanin and total glucosinolates in broccoli sprouts when a pre-harvest Ca application was conducted. In the current study, glucoraphanin content was increased by pre-harvest Zn and Ca treatments, which is in agreement with Yang et al. [54] who found an increase of glucoraphanin content in broccoli sprouts by $ZnSO_4$ and $CaSO_4$ application. The positive role of Ca treatment for increasing glucosinolates might be due to the enhanced biosynthesis of glucosinolates via enhanced *BrST5b* (sulfotransferase 5b) and limited *AOP2* (2-oxoglutarate-dependent dioxygenase 2) expression [55].

**Table 4.** Effect of foliar application of Ca, Mn, Zn, Fe elements plus water as the control on glucoraphanin and glucobrassicin of broccoli florets stored for 28 days at 4 °C.

| | Glucoraphanin (µmol/g FW) | | | | |
|---|---|---|---|---|---|
| **2017** | **Cont.** | **Mn** | **Zn** | **Fe** | **Ca** |
| 0 d | 10.48 ± 0.29 [z] B [y] a | 12.04 ± 0.15 A a | 12.19 ± 0.22 A a | 11.99 ± 0.17 A a | 12.17 ± 0.11 A a |
| 7 d | 9.15 ± 0.04 B b | 11.13 ± 0.10 A b | 11.43 ± 0.24 A b | 11.40 ± 0.03 A b | 11.06 ± 0.12 A b |
| 14 d | 8.36 ± 0.18 C b | 9.14 ± 0.05 B c | 9.166 ± 0.16 B c | 10.18 ± 0.12 A c | 9.11 ± 0.04 B c |
| 21 d | 5.74 ± 0.24 B c | 7.22 ± 0.07 A d | 7.13 ± 0.05 A d | 7.10 ± 0.05 A d | 7.27 ± 0.14 A d |
| 28 d | 4.45 ± 0.21 B d | 6.02 ± 0.05 A e | 6.07 ± 0.08 A e | 5.74 ± 0.15 A e | 5.92 ± 0.07 A e |
| **2018** | Cont. | Mn | Zn | Fe | Ca |
| 0 d | 13.43 ± 0.07 B a | 14.50 ± 0.52 AB a | 15.50 ± 0.58 A a | 14.21 ± 0.26 AB a | 15.08 ± 0.36 AB a |
| 7 d | 12.21 ± 0.09 D b | 13.38 ± 0.08 BC ab | 14.18 ± 0.11 AB ab | 13.01 ± 0.35 CD ab | 14.53 ± 0.09 A a |
| 14 d | 11.76 ± 0.20 C bc | 12.26 ± 0.14 BC b | 13.23 ± 0.00 B bc | 14.37 ± 0.37 A ab | 13.27 ± 0.23 B b |
| 21 d | 9.99 ± 0.21 C c | 10.75 ± 0.14 BC c | 11.84 ± 0.26 A c | 10.76 ± 0.30 ABC bc | 11.39 ± 0.22 AB c |
| 28 d | 6.95 ± 0.55 A d | 8.03 ± 0.32 A d | 8.37 ± 0.17 A d | 8.40 ± 0.37 A c | 8.41 ± 0.20 A d |
| | Glucobrassicin (µmol/g FW) | | | | |
| **2017** | Cont. | Mn | Zn | Fe | Ca |
| 0 d | 4.01 ± 0.01 B a | 5.20 ± 0.01 A a | 4.10 ± 0.05 B a | 5.23 ± 0.03 A a | 4.08 ± 0.04 B a |
| 7 d | 3.91 ± 0.14 A a | 4.11 ± 0.06 A b | 4.09 ± 0.06 A a | 4.10 ± 0.06 A b | 4.07 ± 0.04 A a |
| 14 d | 2.26 ± 0.10 C b | 3.11 ± 0.05 A c | 3.06 ± 0.05 A b | 2.99 ± 0.01 AB c | 2.77 ± 0.04 B b |
| 21 d | 1.40 ± 0.08 B c | 1.91 ± 0.01 A d | 2.00 ± 0.03 A c | 1.97 ± 0.01 A d | 1.85 ± 0.03 A c |
| 28 d | 1.06 ± 0.03 D c | 1.36 ± 0.04 BC e | 1.67 ± 0.05 A d | 1.58 ± 0.05 AB e | 1.23 ± 0.06 CD d |
| **2018** | Cont. | Mn | Zn | Fe | Ca |
| 0 d | 4.39 ± 0.08 C a | 4.94 ± 0.05 B a | 5.44 ± 0.13 A a | 4.99 ± 0.06 B a | 4.89 ± 0.09 C a |
| 7 d | 4.13 ± 0.05 C a | 4.24 ± 0.06 BC b | 4.79 ± 0.07 A ab | 4.546 ± 0.11 AB b | 4.39 ± 0.11 BC a |
| 14 d | 3.06 ± 0.04 B b | 3.11 ± 0.06 B c | 3.67 ± 0.15 A bc | 3.17 ± 0.07 B c | 3.72 ± 0.14 A b |
| 21 d | 2.09 ± 0.08 C c | 2.55 ± 0.11 AB d | 2.95 ± 0.06 A c | 2.35 ± 0.11 BC d | 2.77 ± 0.10 AB c |
| 28 d | 1.54 ± 0.07 B d | 1.83 ± 0.04 A e | 1.95 ± 0.04 A d | 1.77 ± 0.06 AB e | 2.00 ± 0.06 A d |

[z] The results are expressed as mean ± SE of four replicates followed by a letter. [y] Different letters indicate significant statistical differences among fertilizer treatments (upper case) in the same row and days of storage (lower case) in the same column ($p < 0.05$).

Sulforaphane content was decreased during storage periods in both seasons (Table 5). The most effective treatments for increasing sulforaphane content were Ca and Zn. Sulforaphane is a degradation product of glucoraphanin. In our study and the previous work conducted by Šlosár et al. [43], foliar Zn application significantly increased the sulforaphane content in broccoli heads compared with control. Peroxidase activity was increased with increasing storage duration (Table 5). In 2017, broccoli plants treated with Ca, Mn, Zn, and Fe showed the highest values of peroxidase activity compared with control while in 2018 the differences were not significant. It was reported previously that peroxidase activity was enhanced by foliar fertilizer of low Fe dosage in *Lepidium draba* L. (belongs to Brassicaceae family) (seedlings [56]. Flavonoids were decreased with increasing storage periods, as seen in Table 5. All foliar application treatments enhanced flavonoids content in broccoli heads compared with control in both seasons, which is in agreement with Khathutshelo et al. [57] who reported that foliar spray of Fe and Zn increased flavonoids content in tea shoots compared with control.

**Table 5.** Effect of foliar application of Ca, Mn, Zn, Fe elements plus water as the control on sulforaphane, peroxidase and flavonoids of broccoli florets stored for 28 days at 4 °C.

| | | | Sulforaphane (μg/g FW) | | |
|---|---|---|---|---|---|
| **2017** | **Cont.** | **Mn** | **Zn** | **Fe** | **Ca** |
| 0 d | 130.60 ± 0.35 [z] C [y] a | 131.92 ± 0.22 C a | 139.42 ± 0.60 A a | 132.78 ± 0.14 C a | 137.76 ± 1.29 A a |
| 7 d | 104.27 ± 0.65 D b | 108.68 ± 0.51 C b | 113.48 ± 0.80 B b | 106.35 ± 0.54 CD b | 118.89 ± 0.26 A b |
| 14 d | 98.67 ± 0.22 C c | 103.60 ± 0.67 B b | 106.45 ± 0.58 A c | 101.36 ± 0.68 BC c | 106.30 ± 0.42 A b |
| 21 d | 89.55 ± 0.47 A c | 91.00 ± 0.29 A c | 90.96 ± 0.26 A e | 90.77 ± 0.38 A e | 89.31 ± 0.50 A c |
| 28 d | 67.69 ± 0.70 B d | 71.63 ± 0.67 A d | 70.56 ± 0.55 A d | 71.39 ± 0.54 A d | 70.19 ± 0.31 A d |
| **2018** | Cont. | Mn | Zn | Fe | Ca |
| 0 d | 124.78 ± 0.56 AB a | 123.51 ± 0.55 B a | 127.84 ± 1.43 AB a | 125.36 ± 1.15 AB a | 129.38 ± 1.38 A a |
| 7 d | 116.81 ± 0.38 C b | 118.50 ± 0.51 BC b | 120.50 ± 0.58 AB b | 116.13 ± 0.71 C b | 121.45 ± 0.60 A b |
| 14 d | 111.58 ± 0.49 C c | 113.54 ± 0.54 BC c | 115.61 ± 0.57 AB c | 113.50 ± 0.63 BC b | 116.59 ± 0.59 A c |
| 21 d | 99.11 ± 1.75 B d | 101.70 ± 0.61 AB d | 105.40 ± 0.66 A d | 104.42 ± 1.04 A c | 105.25 ± 0.39 A d |
| 28 d | 92.39 ± 1.51 AB e | 90.86 ± 0.35 B e | 95.61 ± 0.43 A e | 94.35 ± 0.99 AB d | 93.43 ± 1.25 AB e |
| | | | Peroxidase (units mg$^{-1}$protein) | | |
| **2017** | Cont. | Mn | Zn | Fe | Ca |
| 0 d | 0.83 ± 0.04 B d | 0.93 ± 0.02 A e | 0.95 ± 0.01 A e | 1.00 ± 0.01 A e | 0.88 ± 0.02 AB e |
| 7 d | 0.96 ± 0.03 C d | 1.74 ± 0.04 A d | 1.74 ± 0.03 A d | 1.79 ± 0.01 A d | 1.47 ± 0.08 B d |
| 14 d | 1.84 ± 0.07 B c | 2.65 ± 0.14 A c | 2.51 ± 0.03 A c | 2.48 ± 0.06 A c | 2.64 ± 0.16 A c |
| 21 d | 3.34 ± 0.06 D b | 5.15 ± 0.07 A b | 4.96 ± 0.09 AB b | 4.71 ± 0.06 B b | 4.24 ± 0.13 C b |
| 28 d | 4.13 ± 0.05 B a | 5.82 ± 0.06 A a | 5.60 ± 0.09 A a | 5.72 ± 0.07 A a | 5.55 ± 0.14 A a |
| **2018** | Cont. | Mn | Zn | Fe | Ca |
| 0 d | 1.01 ± 0.02 B e | 1.19 ± 0.04 AB e | 1.21 ± 0.07 A e | 1.26 ± 0.03 A e | 1.08 ± 0.02 AB e |
| 7 d | 1.66 ± 0.05 B d | 1.85 ± 0.04 AB d | 1.86 ± 0.05 AB d | 1.89 ± 0.04 A d | 1.83 ± 0.04 AB d |
| 14 d | 3.24 ± 0.08 B c | 3.82 ± 0.03 A c | 3.85 ± 0.06 A c | 3.63 ± 0.12 AB c | 3.35 ± 1.10 B c |
| 21 d | 5.43 ± 0.05 B b | 5.67 ± 0.09 AB b | 5.72 ± 0.03 AB b | 5.95 ± 0.04 A b | 5.47 ± 0.11 B b |
| 28 d | 6.25 ± 0.06 A a | 6.72 ± 0.09 A a | 6.63 ± 0.12 A a | 6.64 ± 0.11 A a | 6.43 ± 0.11 A a |
| | | | Flavonoids content (mg/100g FW) | | |
| **2017** | Cont. | Mn | Zn | Fe | Ca |
| 0 d | 101.51 ± 0.68 C a | 120.60 ± 0.48 A a | 116.57 ± 0.50 B a | 119.17 ± 0.57 AB a | 121.40 ± 0.81 A a |
| 7 d | 90.45 ± 0.55 B b | 100.62 ± 0.42 A b | 100.15 ± 0.32 A b | 98.11 ± 0.37 A b | 97.88 ± 1.12 A b |
| 14 d | 76.77 ± 0.55 C c | 84.61 ± 0.55 B c | 88.22 ± 0.33 A c | 85.59 ± 0.41 AB c | 86.98 ± 0.99 AB c |
| 21 d | 57.82 ± 1.05 C d | 69.48 ± 0.54 A d | 70.22 ± 0.98 A d | 71.54 ± 0.67 A d | 64.55 ± 0.47 B d |
| 28 d | 52.44 ± 0.64 C e | 60.33 ± 0.72 A e | 56.33 ± 0.53 B e | 59.61 ± 0.50 AB e | 57.35 ± 1.29 AB e |
| **2018** | Cont. | Mn | Zn | Fe | Ca |
| 0 d | 111.17 ± 1.00 B a | 115.53 ± 0.31 A a | 115.90 ± 1.01 A a | 115.31 ± 0.49 A a | 116.40 ± 0.67 A a |
| 7 d | 101.45 ± 0.57 B b | 105.53 ± 0.52 A b | 104.54 ± 0.55 A b | 107.07 ± 0.64 A b | 106.08 ± 0.63 A bb |
| 14 d | 91.35 ± 0.56 C c | 93.61 ± 0.85 BC c | 97.05 ± 0.34 A c | 95.10 ± 0.37 AB c | 94.65 ± 0.35 AB c |
| 21 d | 81.28 ± 0.60 B d | 83.82 ± 0.21 AB d | 85.73 ± 0.31 A d | 84.80 ± 0.86 A d | 82.42 ± 1.21 AB d |
| 28 d | 67.94 ± 0.93 A e | 69.46 ± 0.55 A e | 69.68 ± 0.56 A e | 69.56 ± 0.38 A e | 66.75 ± 0.79 A e |

[z] The results are expressed as mean ± SE of four replicates followed by a letter.[y] Different letters indicate significant statistical differences among fertilizer treatments (upper case) in the same row and days of storage (lower case) in the same column ($p < 0.05$).

## 4. Conclusions

In this study, we evaluated the effect of supplementary foliar application of Ca, Mn, Zn, and Fe on growth, bioactive compounds, and postharvest behaviour of broccoli heads. Our results showed that plant growth parameters (plant fresh weight, leaf number, head weight, root weight, and head diameter) were not influenced by foliar fertilizer treatments, while SPAD readings were enhanced by Zn. Pre-harvest Ca foliar application showed the lowest weight loss of broccoli heads followed by Mn, Fe, and Zn, respectively. Moreover, Mn and Zn resulted in higher hue angle, lower L* values, and lower chroma values of broccoli heads compared with control treatment. N and P content of broccoli heads were increased by Ca, Mn, Zn, and Fe treatment while K content was not affected.

AsA, crude protein, total chlorophyll content, glucoraphanin, glucobrassicin, sulforaphane, and flavonoids contents were decreased with increasing storage durations while TPC and peroxidase activity was increased by increasing storage periods. Foliar application of Ca, Mn, Zn, and Fe resulted in a higher content of TPC, AsA, total chlorophyll, glucoraphanin, glucobrassicin, flavonoids, and peroxidase activity compared with control plants. Crude protein content was increased by pre-harvest Ca and Mn application when compared with control treatment, while the sulforaphane content of broccoli heads was enhanced by Ca and Zn application. Pre-harvest foliar application of Ca, Mn, Zn, and Fe could be an effective way for enhancing quality, shelf life and preserve bioactive compounds of broccoli heads during refrigerated storage.

**Author Contributions:** "Conceptualization, M.M.E.-M.; Methodology, A.W.M.M.; Software, M.B.I.E.-S.; A.P., and M.M.E.-M.; Formal Analysis, A.W.M.M.; Investigation, M.B.I.E.-S.; Resources, A.P.; Data Curation, M.M.E.-M.; Writing-Original Draft Preparation, M.M.E.-M.; Writing-Review & Editing, A.P.; Visualization, A.W.M.M.; Supervision, M.M.E.-M.; Project Administration, A.W.M.M.; Funding Acquisition, M.M.E.-M.".

**Funding:** This research received no external funding.

**Conflicts of Interest:** The authors declare no conflict of interest.

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
