# Peer review of "Pre-Harvest Foliar Application of Mineral Nutrients to Retard Chlorophyll Degradation and Preserve Bio-Active Compounds in Broccoli"

_agronomy, doi:10.3390/agronomy9110711_

Round 1

Reviewer 1 Report

The subject of the manuscript is consistent with the scope of the Journal. The manuscript is well written, well prepared and focused, providing a full and interesting approach of the effects of the foliar application of minerals on the quality of broccoli, during cold storage.

The authors applied correct analytical methods and received many interesting results, interestingly interpreted that could be disseminated after some minor revisions.

The introduction is well referenced and clearly written.

-L38 a high value crop in terms of what? Rephrase or elaborate this sentence.

-L50 Ca is not a micronutrient. Please correct the term.

The major section that needs editing is the Materials and Methods part. Authors should answer the following:

-How the concentrations of the spraying solutions were selected? Did they trial any preliminary test?

-When the foliar application was applied, at which stage of the plant growth? How many times?

-Where and how the broccoli were stored during the experiment (cold rooms, baskets etc)?

-The authors should provide more information about the methods used. The way they reference the methods needs improvement.

-The section 2.4.4 needs to be re-written. It is confusing. What do you mean by “several content”? Is it serial dilution? Authors should check the volumes and the units illustrated here as well. Also, it is PAD and not DAD.

-Also, lines 122-123, needs to be rephrased.

-Why the authors did not assess the hardness of the tissue (texture analysis) or the marketability of the final product after storage. Any comments on the general appearance could be useful.

Regarding the Results and Discussions part, It would be easier for the reader to see in graphs and in tables the results of the CONTROL treatments first, before the applications.

-I would also suggest that the results of weight loss, and all of Figure 2, to be illustrated as bars, in way for the statistical analysis to be more evident (and displayed, it is missing now on these figures).

-Table 1. Use for macronutrients g/kg and for micronutrients mg/kg, as units.

Change units at abstract as well. The ‘’ppm’’ is not appropriate, as scientific community prefers other formats

-Page 9 L6-8, please be more specific here, providing the point you want to make.

-Also you can connect the SPAD results with the chlorophyll content results.

L120. Check units format

L129 and elsewhere. Use common way for unit’s presentations according to the journal author guidelines. For example ml or mL? μl or μL??

L198. Provide the names of the citation, or otherwise move them at the end of the sentence

Table 2. Please provide the units of TPC as (mg GAE/g FW)

Author Response

Reviewer one

Dear Reviewer,

Thank you very much for your valuable comments. We have carefully reviewed the manuscript in line with the comments you made and those of the reviewers and respond below point by point to the comments made (markerd as Author Answer – AA).

-L38 a high value crop in terms of what? Rephrase or elaborate this sentence.

AA: The correction has been carried out. It is stated now that broccoli is a high commercial value crop.

-L50 Ca is not a micronutrient. Please correct the term.

AA: The correction has been carried out, through out the manuscript, also in the title micronutrient has been replaced with mineral nutrient.

The major section that needs editing is the Materials and Methods part. Authors should answer the following:

-How the concentrations of the spraying solutions were selected? Did they trial any preliminary test?

AA: The corrections have been chosen depending on preliminary test of these products. The correction has been carried out, please refer to line 87-88.  

-When the foliar application was applied, at which stage of the plant growth? How many times?

AA: We have mentioned these the days after transplanting in lines 88-89, (after 30 days of transplantation).

-Where and how the broccoli were stored during the experiment (cold rooms, baskets etc)?

AA: The correction has been carried out, refer to section 2.3.

-The authors should provide more information about the methods used. The way they reference the methods needs improvement.

AA: According to Agronomy journal style, they mention that new methods and protocols should be described in detail while well-established methods can be briefly described and appropriately cited. Thus, most of our measurements well-established such as N, P, K, Vit C, Ca, Zn, Fe, Cu, total chlorophylls, phenolic contents, and total flavonoids. However, we described in details the other parameters such as sulforaphane, glucosinolates, and peroxidase activity.

-The section 2.4.4 needs to be re-written. It is confusing. What do you mean by “several content”? Is it serial dilution? Authors should check the volumes and the units illustrated here as well. Also, it is PAD and not DAD.

AA: Modification made in the section as suggested by the reviewer, please refer to the section 2.4.4 for further details.

-Also, lines 122-123, needs to be rephrased.

AA: Changes made as suggested by the review, please refer to section 2.4.5, line 143-145

-Why the authors did not assess the hardness of the tissue (texture analysis) or the marketability of the final product after storage. Any comments on the general appearance could be useful.

AA: Thank you for your useful comment. We already tested the surface color which gave us accurate results better than general appearance by naked eyes. Also, according to firmness, we cannot measure the heads because it is lose and the stem is not important for measure.

Regarding the Results and Discussions part, it would be easier for the reader to see in graphs and in tables the results of the CONTROL treatments first, before the applications.

AA: The correction has been carried out.

-I would also suggest that the results of weight loss, and all of Figure 2, to be illustrated as bars, in way for the statistical analysis to be more evident (and displayed, it is missing now on these figures).

AA: Thank you for your comment. However, we believe that the periodical data should be presented as it is.

-Table 1. Use for macronutrients g/kg and for micronutrients mg/kg, as units.

Change units at abstract as well. The ‘’ppm’’ is not appropriate, as scientific community prefers other formats.

AA: ppm is converted into mg/kg as suggested by the reviewer.

-Page 9 L6-8, please be more specific here, providing the point you want to make.

AA: Now at Page 11, line 8-9 , modified as suggested by the reviewer.

-Also you can connect the SPAD results with the chlorophyll content results.

AA: Yes, that’s right SPAD and Chlorophyll content results can be connected in certain cases where SPAD data is required to calculate chlorophyll content, a correlation function can be used. However, in our case the objective of the study was not to correlate SPAD and Chlorophyll content. As this will add more statistical analysis and results will be slightly far from our primary objectives, we would rather avoid doing this in this paper.

L120. Check units format

AA: The correction has been carried out.

L129 and elsewhere. Use common way for unit’s presentations according to the journal author guidelines. For example ml or mL? μl or μL??

AA: The correction has been carried out. The author guidelines were followed.

L198. Provide the names of the citation, or otherwise move them at the end of the sentence

AA: The correction has been carried out.

Table 2. Please provide the units of TPC as (mg GAE/g FW)

AA: The correction has been carried out.

Reviewer 2 Report

Comments:

There is no description of soil conditions in which broccoli was grown. Please describe and provide climatic conditions (precipitation, temperature) during the broccoli growing season. Lack of confirmation of the reliability of analytical methods - certified material. Other remarks were made in the text.

Author Response

Review two

There is no description of soil conditions in which broccoli was grown. Please describe and provide climatic conditions (precipitation, temperature) during the broccoli growing season. Lack of confirmation of the reliability of analytical methods - certified material. Other remarks were made in the text.

Reply to the reviewer:

Thank you very much for your useful comments. Soil composition (type) and climatic condition during the broccoli growing season has been incorporated in the text as required by the reviewer (Please refer to Section 2.1, Line 73 -80).

The mineral composition analysis method (Section 2.4.2) has been modified as highlighted by the reviewer, to provide reliability of the certified analytical method.

All other minor remarks made in the text has been addressed by the author, please refer to the track changes in the manuscript.  
